# A Mindfulness-Based Mobile Application’s Impact on Nurse Burnout Syndrome and Well-Being

**DOI:** 10.3390/healthcare13192386

**Published:** 2025-09-23

**Authors:** Jennifer Wedster, Jennifer DiBenedetto

**Affiliations:** 1Department of Nursing, Trinity Christian College, 6601 W College Dr., Palos Heights, IL 60463, USA; 2The Richard and Sheila Young School of Nursing, Regis College, 235 Wellesley St., Weston, MA 02493, USA

**Keywords:** nurses, emergency department, burnout, stress, well-being, mindfulness, mindfulness-based mobile applications, Moodfit

## Abstract

**Background/Objectives**: Burnout syndrome among nurses can significantly contribute to the nursing shortage, leading to high turnover and negative impacts on both nurses and patient care. The primary objective of this project was to evaluate the feasibility, acceptability, and preliminary effect of a mindfulness-based mobile application (MBMA) on burnout and well-being in emergency department (ED) nurses over four weeks. **Methods:** An EBPQI with a descriptive approach was taken to evaluate ED nurses’ burnout and well-being, which was measured with the Mini-Z Single Item (MZSI) and Nurses’ Well-Being Index (NWBI). We also asked three open-ended questions about their experience using the once-daily MBMA over the four-week period. Twelve participants from a mid-western hospital were recruited, and six completed both the pre-test and post-test surveys. **Results:** Results found no statistically significant improvement in burnout (*p* = 1.00) or well-being (*p* = 0.783). However, upon a secondary analysis using imputed data, a statistically significant improvement in burnout was found (*p* = 0.012). Among the six participants who completed the post-intervention, a significant and positive correlation between burnout and well-being was identified (*r* = 0.81, *p* = 0.048). Themes from qualitative responses included perceived helpfulness of MBMA tools, perceived usefulness, and lack of time for daily participation. Although statistical improvements were not observed, individual comments indicated that the tool was helpful; however, setting aside time to engage with it remained difficult. **Conclusions:** Findings from this project support the need for further research exploring the impact of individualized interventions specifically targeting ED nurses as well as organizational strategies aimed at those already experiencing burnout or impaired well-being.

## 1. Introduction

Nursing is considered one of the most stressful professions in healthcare [1,2]. This increase in stress is often attributed to the high environmental stressors that nurses experience, such as long work hours, shift work, increased physical and emotional demands of nursing, constant exposure to ethical dilemmas, increased demands from the aging population, and sustained inadequate workforce, which all contribute to the development of burnout [1,3,4]. Additionally, other factors contributing to work stress, such as inadequate workforce support (e.g., under-enrollment in nursing programs), the current workforce nearing retirement, decreases in job satisfaction, and the COVID-19 pandemic, are known to exacerbate these systematic environmental stressors [5,6]. When these environmental stressors are prolonged and in excess, it can position nurses to experience a phenomenon called burnout syndrome [2]. Burnout syndrome is a condition that manifests through at least one of three dimensions: a heightened sense of physical or emotional exhaustion (EE), the emergence of feelings of disconnection from one’s work or depersonalization (DP) typically stemming from EE, or a diminished sense of personal accomplishment (PA) [7,8]. Burnout syndrome has been shown to have negative implications affecting the nurse, patient, and healthcare organizations. Interventions such as mindfulness-based mobile applications may be one strategy to reduce feelings of burnout and promote well-being among nurses employed in healthcare settings.

## 2. Background

Nurses, patients, and healthcare organizations can often experience some of the many negative implications stemming from burnout syndrome. Among Registered Nurses (RN), implications from burnout syndrome include various ailments, including being at an increased risk for suicide [9,10]. Stressful work environments, rotating shifts, constant exposure to difficult emotional situations, and communicable diseases can further increase the risk of mental ailments among RNs [11,12]. Nurses with burnout syndrome were also found to have a decreased commitment to the organization and an increased intent to leave their job positions [13], with as many as 41–47% citing stress and burnout as contributing factors [14,15]. Among all nursing specialties, nurses employed in the Emergency Department (ED) were found to have the highest risk of acquiring burnout syndrome [1]. This increased risk was attributed to ED nurses working in high-stress and high-paced environments, thereby increasing their risk for turnover and burnout [1]. One study found that ED nurses were 13 times more likely to consider leaving their place of employment when experiencing high emotional exhaustion, a major component within burnout syndrome [16].

When these nurses experience the negative implications stemming from burnout syndrome, it can simultaneously and negatively impact patient care. The literature suggests that patients were at an increased risk for poor health outcomes when the nurses who cared for them experienced burnout syndrome [6,13]. Some of these poor health outcomes included increased infection rates and nurses’ non-adherence to medication administration safe practices [13]. Specifically, within the distinct domains comprising burnout syndrome (EE, DP, and PA), nurses who experienced an increase in EE increased their patients’ risk for mortality, while those experiencing DP placed patients at a greater risk of experiencing falls and medication errors [17].

Burnout syndrome in nursing not only impacts patient care but also leads to nurses leaving the bedside, resulting in increased costs for healthcare organizations [18]. One of the largest healthcare expenditures is nurse turnover [19]. In a recent survey by Nursing Solutions (2023) [19], authors estimated the financial cost to replace a nurse to be USD 52,000 per nurse if they leave within the first year; as nursing turnover continues, organizations are at risk for losing millions of dollars over time. Therefore, it is recommended for healthcare organizations to consider investing in mental health support for nurses experiencing burnout syndrome to ensure their well-being and effectiveness and reduce nurse turnover [12].

One effective strategy that has been shown to reduce burnout syndrome and enhance physical and emotional well-being among nurses includes mindfulness-based interventions. Mindfulness is defined as the intentional act of being aware of the present moment to support the mind and body, avoid letting the mind wonder off, and find balance, reduce stress, and support well-being in a non-judgmental way [20,21,22]. Numerous studies suggest that when individuals engage in mindfulness-based interventions (MBIs) it can significantly improve their physical and mental well-being and reduce burnout [23,24,25,26,27]. Within the healthcare sector, several studies support the use of mindfulness-based mobile applications (MBMAs) as an effective strategy to facilitate mindfulness practices and improve well-being. One study by Knill et al. (2021) [28] found that nurses who worked in oncology and engaged in MBMAs were found to be effective in reducing the risk of burnout and improving their well-being. Similarly, among neonatal intensive care unit nurses, nurses who engaged in MBMAs were found to have significant improvements in their overall quality of life [29].

Mitigating burnout may improve nurses’ well-being, promote quality patient care, and potentially decrease the fiscal burden on organizations from nurse turnover. While various healthcare associations, including the American Nurses Association (2024) and the National Academy of Medicine (2023), support healthcare organizations in targeting improving the well-being and burnout of the healthcare workforce through MBIs, a practice gap remains in its application in the clinical setting, like the ED [30,31]. The purpose of this evidence-based practice and quality improvement (EBPQI) project was to explore the feasibility and acceptability for ED nurses to use a once daily MBMA (Moodfit^TM^) tool and its preliminary effect on their feelings of burnout and well-being over a four-week period.

## 3. Materials and Methods

An EBPQI project utilizing a pre-test and post-test descriptive design was conducted and was grounded in the Johns Hopkins Evidence-Based Practice Model framework. We recruited nurses from one local ED in Illinois, USA, after explaining and defining mindfulness and MBMAs and what would be expected of them. To participate in the project, we required ED nurses to meet the following inclusion criteria: (1) be a RN working in the ED either registry, per-diem, full-time, part-time, or travel; (2) have access to a smartphone; be willing to use a MBMA for 10 min daily for a period of four weeks; and (3) possess the necessary technology skills to download and utilize a mobile application. We excluded nurses who did work in the ED or had a job role outside of being a RN. Over Qualtrics (Qualtrics.com, Provo, UT, USA), we provided an informed consent document for respondents to review the project information and provide consent. Participants then completed the Mini-Z Single Item (MZSI) and the Nurses’ Well-Being Index (NWBI) surveys to measure their baseline burnout and well-being, respectively, followed by daily engagement with a MBMA called Moodfit^TM^ for ten minutes daily over a four-week period. After four weeks, we administered the same burnout and well-being surveys to the ED nurses while also asking three open-ended questions about their experience after using the MBMA. To maintain confidentiality and anonymity of participants, we ensured all collected data was de-identified, analyzed in aggregate, and stored securely on a password-protected computer. We obtained Institutional Review Board approval from both Regis College and the partnering organization to conduct this project.

### 3.1. Measurements

To obtain participant data to explore the baseline levels of burnout and well-being, we first collected demographic information about the participants. Although not all participants remained in the project, their data was kept to compare characteristics of those more likely to be retained and engaged. The demographic information we collected included age, gender, ethnicity, race, level of education (LOE), employment status, years of experience as a nurse, years of experience as an ED nurse, and presence of a disability or chronic condition(s). Primary data were analyzed using the Wilcoxon Signed-Rank test, comparing pre- and post-baseline differences among the burnout and well-being scores. A secondary analysis, using the imputation process, was also conducted to account for incomplete data and not ignore baseline levels of burnout and well-being. To measure changes in burnout and well-being, we administered the Mini-Z Single Item (MZSI) and the Nurses’ Well-Being Index (NWBI) surveys before and after they engaged in the MBMA. Additionally, a Pearson’s Correlation Test was completed to evaluate the relationship between participants’ level of burnout and well-being pre-intervention. Then we explored whether there was a correlation between burnout and well-being post-intervention. At the end of the project, we asked respondents to answer three additional open-ended questions to capture their experience using the MBMA to help modify and make improvements to the current practice change and provide recommendations to the participating healthcare organization.

### 3.2. Instruments

All instruments used in this project to measure burnout and well-being were found to be valid and reliable. The MZSI, comprising a single item with a 5-point Likert scale, was found to be an effective scale endorsed by the National Academy of Medicine due to its fast completion and correlation with the emotional exhaustion domain within the Maslach Burnout Inventory (*r*^2^ = 0.63–0.65, *p* < 0.05) [32,33,34]. The MZSI was chosen because we hoped it could potentially increase response rates among healthcare providers, where emotional exhaustion is the primary domain of interest (*r*^2^ = 0.5, *p* < 0.0001) [34]. Participants who engaged with the survey were asked to rate their feelings at that moment on a scale from 1 to 5. A score of “1” indicated “no symptoms of burnout,” while scores of “3” and above suggested the presence of burnout [33]. A score of “5” meant “complete burnout and a need for additional help” [33].

To measure nurses’ well-being, we administered the nine-item NWBI instrument. This survey is an adapted version of the Well-Being Index to screen specifically for RN distress and measure their overall well-being [32,35]. Among the nine items, the instrument asked five dichotomous (yes/no) responses while asking two additional Likert scale questions to evaluate burnout, fatigue, low mental/physical quality of life, depression, and anxiety/stress [32,35]. The first seven items are awarded a +1 if answered with a yes, and the last two items (Likert scale items) are assigned a −1, 0, or +1 depending on the response. The score for the NWBI ranges from −2 (lowest risk) to 9 (highest risk), with a score equal to or greater than 2 indicating a greater risk for adverse outcomes, including burnout, fatigue, poor quality of life, patient care errors, and intent to leave within the next two years [36]. The WBI has been validated in multiple studies and confirmed reliable by intra-rater reliability processes, which included over 25,800 healthcare professionals, yet the nurses’ version has not had reliability and validity testing [35].

Given that we also wanted to capture the nurses’ overall experience with using MBMAs and engaging in mindfulness practices, three open-ended, descriptive questions were collected after they engaged in the MBMA intervention. The questions focused on identifying any challenges during MBMA use, determining which activity they found most useful, and identifying why they would or would not keep using the MBMA. Themes were developed from the questionnaire to gain insight into how they felt when engaging in the MBMA and for us to explore their experiences with their responses from the burnout and well-being survey data.

### 3.3. Intervention

To participate in the MBMA intervention, we asked participants to download the Moodfit 30-day free trial on their personal IOS cellular devices. We provided participants with two options to engage in the Moodfit MBMA: either to utilize the MBMA for at least ten minutes by listening to a meditation recording or participate in at least one mindfulness activity within the application once daily over four weeks. We opted to provide the ED nurses a flexible mindfulness choice selection intentionally to promote compliance and retention, since they may have unique preferences for engaging in mindfulness. Some of the Moodfit MBMA’s choice exercises include meditation, breathing exercises, mood tracking, coping strategies, gratitude journaling, or audio-guided meditation to improve awareness of one’s mood and sort through thoughts and feelings [37]. The application supports ease of use with an easy-to-navigate home page featuring users’ progress and goals, and the option to customize their goals and preferred tools, such as the reminder tool and progress on the home page.

## 4. Results

While 12 participants completed the pre-tests, only 6 completed the post-tests. Frequencies and percentages were calculated based on the survey data and analyzed in Intellectus360 (statistics.intellectus360.com, Clearwater, FL, USA) alongside the assistance of a statistician. Due to the low sample size, we ran a Wilcoxon Signed-Rank Test to compare preliminary differences in burnout and well-being among the 6 participants who completed the study. To account for all 12 participants who completed the pre-test, a secondary analysis using the imputation process was conducted on the 6 participants who did not complete the post-test MZSI and NWBI surveys. A Pearson Correlation Test was conducted to investigate whether associations or relationships between burnout and well-being existed among participants, particularly in those who were retained fully in this project.

### 4.1. Demographic Results

Within the 12 participants who consented and completed the demographic survey, a subsample 33% (n = 11) were between the ages of 40 and 49, 92% were female (n = 11), 59% were White or Caucasian (n = 7), and 75% were not Hispanic or Latino (n = 9). Educationally, half of the participants held a master’s degree (e.g., MA, MS, MSN) (n = 6). In terms of employment, 75% were employed full-time (n = 9), half had greater than 9 years working as a RN (n = 6), and 32% had greater than 9 years of experience working in the ED (n = 4). No participants identified as having a disability or chronic disease (Table 1).

Of the six participants who remained in the project and completed the demographic survey, 67% (n = 4) were between 40 and 60 years of age. All participants identified as female (100%, n = 6), and 83% identified as White or Caucasian (n = 5) and not Hispanic or Latino (n = 5). Educationally, 83% held a master’s degree (e.g., MA, MS, MSN) (n = 5). Regarding employment status, 67% were employed full-time (n = 4), had 7 or more years of RN experience (n = 4), and had 7 or more years of experience working in the ED (n = 4).

### 4.2. Mini-Z Single Item Survey

Among ED nurses who completed the pre-test MZSI and were retained at post-test, 67% (n = 6) reported having emotional exhaustion (burnout) at pre-test, which was characterized by selecting “one or more symptoms of burnout,” or “having symptoms that will not go away” on the MZSI. The remaining 33% of participants (n = 2) did not report having any burnout, yet they selected having some feelings of stress.

Similar to pre-test findings, out of the six participants who fully completed the four-week MBMA intervention and MZSI survey, 67% (n = 6) continued to report emotional exhaustion or “having symptoms that will not go away.” The remaining 33% did not meet the criteria for burnout and only reported having stress. Table 2 provides a breakdown of these scores in aggregate against the single-item questionnaire.

When primarily analyzing responses from the six participants who completed both the pre- and post-test surveys completely, the Wilcoxon Signed-Rank Test did not have significant findings (Z = 0.00, *p* = 1.00). Despite these insignificant findings, individual scores demonstrated some variation, evident in Figure 1, which presents a profile plot highlighting all six participant scores before and after completing the MZSI.

Upon running a secondary analysis on the 12 participants who completed the pre-test, frequency data showed that 33% (n = 4) of participants did not meet the clinical cut off for burnout requirements, with 8% (n = 1) of participants reporting no feelings of burnout and 25% (n = 3) reporting being under stress but not feeling symptoms of burnout. Among participants who met the burnout criteria, 50% (n = 6) reported feeling burnout and 17% (n = 2) reported having symptoms of burnout that would not go away. Due to half of the participants falling to attrition at post-test, an accurate assessment of their combined burnout score cannot be displayed definitively since the imputed data cannot accurately depict meaning to a single-item response.

To control for this attrition, a Wilcoxon Signed-Rank Test was run using imputed data to account for the six missing responses in the post-test. Results were found to have statistically significant differences in ED nurses’ burnout score (Z = −2.51, *p* = 0.012). Table 3 presents the findings of the Wilcoxon Signed-Rank Test on the imputed data.

### 4.3. Nurses’ Well-Being Index Survey

When primarily analyzing the 12 participants who completed the pre-intervention surveys, 25% (n = 3) of participants did not meet criteria for impaired well-being, 33% (n = 4) of participants met criteria for being at risk of impaired well-being, and 41% (n = 5) met criteria for impaired well-being. After engaging in the four weeks of the MBMA, only half of the participants responded (n = 6). A total of 17% (n = 1) did not meet criteria for impaired well-being, 50% (n = 3) met criteria for being at risk for impaired well-being, and 34% (n = 2) met criteria for experiencing impaired well-being. Figure 2 presents a profile plot showing each of the six participants before and after NWBI scores.

A Wilcoxon Signed-Rank Test of the pre-test and post-test NWBI scores of the six participants who completed both surveys fully was found to be insignificant (Z = −0.28, *p* = 0.783) (Table 4).

After running a secondary analysis on the 12 participants who fully completed the pre-test NWBI survey against the 6 participants who fully completed the post-test NWBI and accounting for the 6 participants who fell to attrition using the imputation process, results from the Wilcoxon Signed-Rank Test found insignificant differences in well-being after engaging in the four-week MBMA (Table 5).

### 4.4. Pearson Correlation Between Burnout and Well-Being Scores

When we investigated the correlation between the pre-test burnout and well-being scores among the six participants who completed all surveys, no significant correlation was observed (90% CI = [−0.73, 0.75], *r* = 0.03, *p* = 0.955). Figure 3 presents the scatterplot with a regression line added for the pre-burnout and well-being scores. Appendix A can be accessed and provides the raw data analysis demonstrating the Pearson Correlation analysis between burnout and well-being [38,39,40]. 

However, when we investigated the post-test correlation between the six ED nurses’ responses to the NWBI and MZSI scores after engaging in the MBMA, a significant, moderate, and positive correlation was observed (90% CI = [0.19, 0.97], *r* = 0.81, *p* = 0.048). Figure 4 presents the scatterplot with a regression line added for the post-burnout and well-being scores.

### 4.5. Thematic Analysis

Six participants completed the open-ended questions at the completion of the four-week MBMA intervention. Three themes emerged among the ED nurses: (1) perceived helpfulness of the MBMA tools, (2) perceived usefulness of the MBMA, and (3) lack of time.

#### 4.5.1. Theme 1: Perceived Helpfulness of the MBMA Tools

Participants identified various tools within the Moodfit^TM^ MBMA that were either helpful and/or challenging for them to use over the four-week period. Four of the six participants outlined the most helpful aspects of the MBMA, particularly the daily text reminders and mood tracker. Participants stated, “*The daily reminder to pause and breathe, along with my trending results, are helpful,*” “*It gave me practical recommendations to improve my mood,*” “*I like the daily text reminders that I set up,*” and “*I found the daily reminders and mood selections to be the easiest to use*.” The option to select different types of mindfulness therapies was appealing to one participant, who reported, “*Yes, it helped to get everything out and all the options for ways to cope and different therapies were so beneficial.*”

Regarding the overall helpfulness of the MBMA to their well-being, some participants reported the MBMA’s benefits. One participant exclaimed, “*[The MBMA] did help in some ways. It allowed me to reflect on things and people that I am grateful for…*” while another reported, “*It was helpful…I found it good overall. I like the points of possible success and joy that it allowed.*”

Despite four participants sharing positive feelings about how helpful the MBMA’s tools are, two remaining participants expressed challenges or dislikes when using them. These concerns involved difficulties with editing and the limited tools within the MBMA, evident in the following quotes: “*It was hard to edit items*,” and “*I didn’t like how there weren’t more prompts.*”

#### 4.5.2. Theme 2: Perceived Usefulness of the MBMA

Different degrees of usefulness were also observed among participants regarding the MBMA. Participants who reported actively using it found it beneficial, but a few were uncertain about its continued use. When evaluating continued use of the MBMA after the project, one participant, who reported daily use of the MBMA through the reminders tool, stated, “*Yes… it improved my well-being,*” and “*I felt it was impactful and easy to use*.” However, a couple of participants, while reporting active usage, were somewhat uncertain about its use after the four-week period. Some participants stated, “*I think I will use [the MBMA]”* and “*I am considering it.*”

#### 4.5.3. Theme 3: Lack of Time

Some participants noted a lack of time to fully engage and participate in the MBMA, while emphasizing they will likely not continue using the MBMA after the project. Although one participant reported no “*problems with the app*,” they followed up by stating, “*It took time to perform the process which is in short supply as it is*.” Other participants felt similarly and exclaimed, “*I would find myself not making time to utilize it*,” “*It took time to perform the process…it’s just one more thing to add to the ever-growing list of responsibilities*,” and “*I felt at times it was something I was just checking off a list*.”

## 5. Discussion

Our EBPQI project evaluated the feasibility, acceptability, and preliminary effect of a 10 min daily MBMA Moodfit^TM^ on burnout and well-being in ED nurses over a four-week period. Our primary outcome was not to draw causal conclusions; rather, it was to explore the feasibility and acceptability of ED nurses engaging in the MBMA utility in ED nursing practice. Results in this project found significant differences in burnout scores among the 12 participants who completed the pre-test with the imputed scores at post-test. However, no statistically significant differences in burnout among the six participants who fully completed both surveys at pre-test and post-test, or well-being, were found in both the primary and secondary analyses. Despite these mostly insignificant findings, we hypothesize, based on their qualitative responses against the survey responses, that the participants who were retained and engaged with the MBMA demonstrated improvements in well-being and found the reminder tool helpful. Conversely, ED nurses who reported less engagement in the MBMA had either no change or worsening levels of burnout and well-being. Therefore, we recommend researchers encourage participants to use the reminder tool on their mobile devices to foster engagement with the MBMA. These recommendations align with the literature demonstrating improvements in burnout and well-being when ED nurses are fully engaged. Results from a two-armed RCT showcased how ED staff who engaged with a MBMA (Headspace^TM^) had significant improvements in burnout and well-being [26,27]. Similarly, an EBPQI article identified a significant decrease in burnout after cardiac nurses practiced 10 min of meditation exercises using the MBMA (Headspace^TM^) [41]. Additionally, a study evaluating the relationship between mindfulness and subjective well-being discovered that those practicing mindfulness reported higher levels of well-being among participants selected from the general population [42]. These findings support the benefits of practicing mindfulness and, when engaged, the potential to improve an individuals’ sense of overall well-being. Conversely, other studies showed the positive impact MBMAs had on improving symptoms of depression (Lift^TM^) and post-traumatic stress disorder (Moodfit^TM^), but the burnout domain remained insignificant among nurses working in high acuity environments [29,43].

Our findings observed a moderate and positive correlation between having poor well-being and worsening burnout. Although we cannot fully determine the correlation due to the small sample size, it may be attributed to the ED nurses becoming more mindful of their feelings of burnout and poor well-being than before engaging with the MBMA, particularly since mindfulness is designed to promote awareness and focus on their emotions. Additionally, the ED nurses in our project reported experiencing high levels of stress, burnout, and impaired well-being at baseline, which is very likely one of the reasons why non-compliance was an issue in this project. While there was a high attrition rate, nearly half of the retained participants reported not having enough time to engage in the MBMA intervention. This perception of the “lack of time” and “just another task to complete” to fully engage in the daily mindfulness activities, even for ten minutes, was unsurprising and consistent with the literature [43]. In one qualitative study evaluating the Calm^TM^ MBMA among nursing staff, the authors found similar barriers to usage, such as a “lack of time” and “forgetting to use it.” However, participants in the study who used the MBMA daily also found it helpful in reducing stress and feeling calm [44]. Although we did not measure compliance directly in this project, we asked questions about their experience using the MBMA to capture similar compliance data.

Another interesting finding was that among the ED nurses who engaged in the MBMA, nurses with graduate degrees (specifically at the MSN level) completed the intervention, with one additional participant’s level of education at the BSN level. While the sample size was very small, it could be hypothesized that nurses with higher levels of education are more likely to seek and utilize individual-level resources and engage in research-related projects compared to those with lower levels of education. This hypothesis is supported by past literature involving a range of healthcare workers who were recruited to practice a MBI, with most participants reporting education at the graduate level or higher [45]. Additionally, when MBMAs between nurses and nurse practitioners was employed on a bone marrow transplant unit, researchers noted higher attrition rates among RNs when compared to nurse practitioners [28]. However, high attrition rates (50%) were found to be consistent with other studies in other EBPQI projects that integrated MBMA interventions across job roles and levels of education [29,43]. This finding emphasizes the need for additional resources to support the retention of nurses who hold a BSN or a lower degree. It is also important to consider that the project was implemented towards the end of the influenza season; when the patient census was high, the unit was often short-staffed, and travel nurses were frequently used.

Improving well-being or addressing burnout is not a shortcoming of the individual; rather, it is a system and organizational issue that cannot be addressed solely at the individual level. While using a MBMA may help mitigate burnout and improve well-being for participants motivated to use the MBMA, we emphasize the importance of considering implementing MBMA projects to nurses experiencing less burnout at baseline and using it as a resilience-building strategy after organizational-level burnout-reducing interventions occur. Future research can consider exploring these factors when implementing similar MBMA interventions in nursing departments that are most vulnerable to experiencing higher burnout and poorer well-being.

### Limitations

We recognize that this EBPQI project has several limitations that may affect the interpretation of the project outcomes. The primary limitation of this study is the small sample size of six participants, which increases sensitivity to outliers and restricts the generalizability of the results to a larger population. We tried to account for the very small sample size by running a secondary analysis of the 12 participants who partially completed the project. While imputed data allow us to account for pre-intervention burnout and well-being levels, they cannot replace actual measurements, and we cannot assume those who did not complete the post-test surveys would have had the same outcomes. The high attrition rate could also lead to potential sample bias, as the remaining participants may not accurately represent the original group of 12 participants. Additionally, the sample was limited to a small sample of ED registered nurses, which limits its transferability to other nursing specialties or drawing causal conclusions of the MBMAs’ impact on burnout and well-being. Ultimately, the loss of data and small sample size affect the validity and reliability of the findings, reducing the power of the analysis and making it difficult to establish consistent patterns. Despite these limitations, meaningful insights into the acceptability and feasibility of the project findings can be determined while providing interesting preliminary results for further investigation in additional EBPQI projects and/or research.

## 6. Conclusions

Nurses employed in the ED work in high-stress environments that impose significant physical and psychological demands. These nurses need resources to mitigate burnout and support their overall well-being. Although the MBMA did not generate any significant improvements in well-being and only suggested significant differences in burnout with imputed data, the open-ended responses suggest MBMAs may enhance individuals’ awareness of their own levels of burnout and well-being. These outcomes, combined with existing literature on this topic, suggest that implementing an MBMA in nursing practice may serve as a beneficial individual-level resource for ED nurses. However, there remains a critical need to integrate interventions specifically aimed at supporting the well-being of ED nurses already experiencing high levels of burnout or impaired well-being. Future, high-quality projects and research are needed to establish consistent trends and long-term outcomes to promote wellness in nurses.

## Figures and Tables

**Figure 1 healthcare-13-02386-f001:**
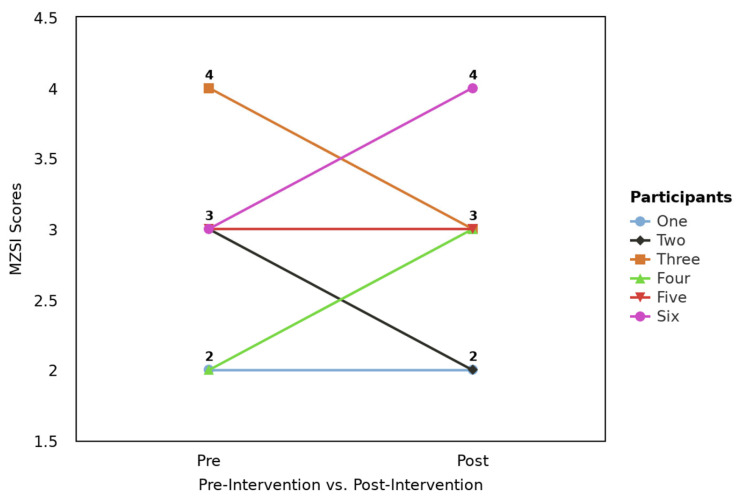
Profile plot for pre- and post-Mini-Z Single Item (MZSI) scores by the six participants who completed the pre-test and post-test survey.

**Figure 2 healthcare-13-02386-f002:**
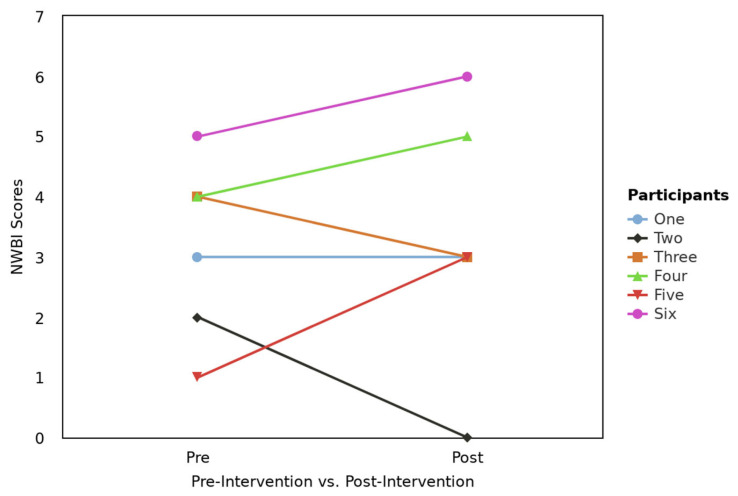
Profile plot for pre- and post-NWBI scores by the six participants who completed the pre-test and post-test survey.

**Figure 3 healthcare-13-02386-f003:**
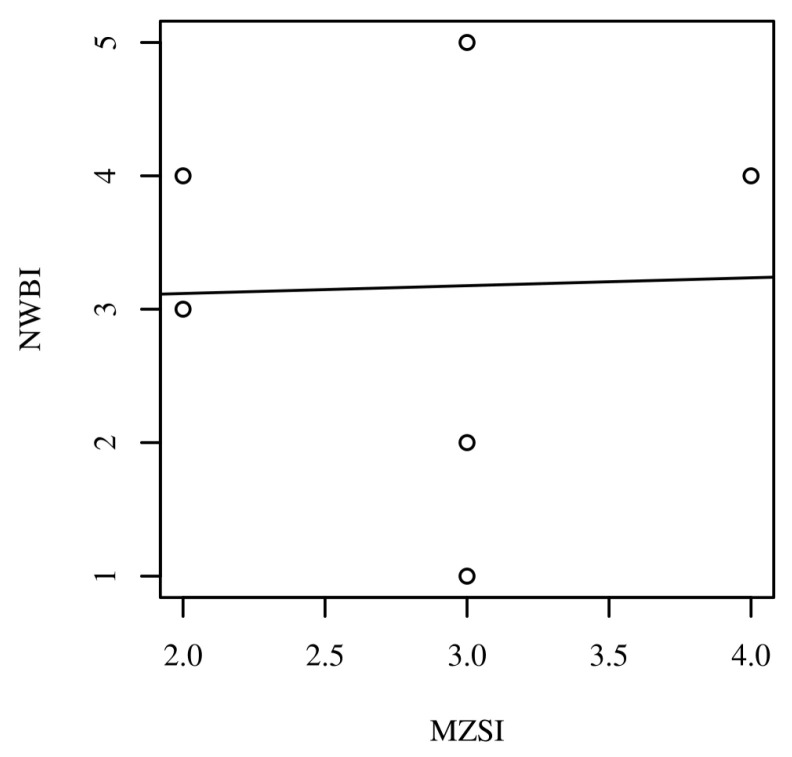
Scatterplots with regression line added for the pre-intervention Mini-Z Single Item (MZSI) and Nurses’ Well-Being Index (NWBI) scores for the six participants who completed both the pre- and post-test surveys.

**Figure 4 healthcare-13-02386-f004:**
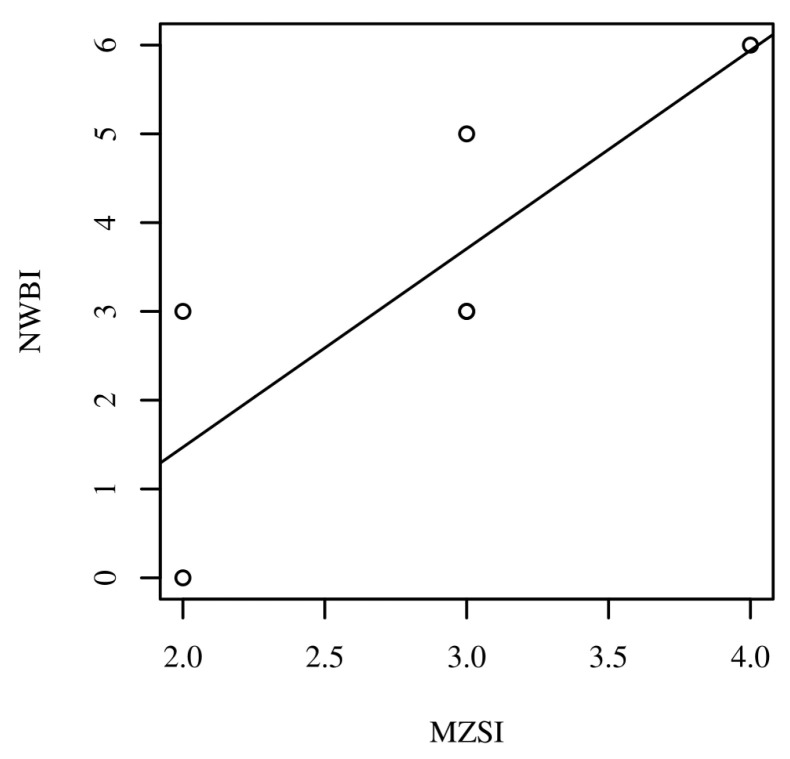
Scatterplots with regression line added for the post-intervention Mini-Z Single Item (MZSI) and Nurses’ Well-Being Index (NWBI) scores for the six participants who completed both the pre-test and post-test surveys. Note: Some scores overlap and may not adequately show the number of participants between each correlation observation, as two participants scored a three on both the MZSI and NWBI surveys.

**Table 1 healthcare-13-02386-t001:** Descriptive statistics for participants’ demographic variables at baseline. Due to rounding errors, percentages may not equal 100%. * Other is not specific to maintain confidentiality.

Variable (N = 12)	n	*%*
Age		
20–29	2	17
30–39	3	25
40–49	4	33
50–59	3	25
Gender		
Female	11	92
Male	1	8
Race		
White or Caucasian	7	59
Black or African American	4	33
American Indian or Alaskan native	1	8
Ethnicity		
Hispanic or Latino	1	8
Not Hispanic or Latino	9	75
Other *	2	17
Employment Status		
Employed Full-time	9	75
Registry or Per diem	2	17
Employed Part-time	1	8
Presence of a Disability or Chronic Disease		
No	12	100
Number of Years as RN		
3–5 years	1	8
5–7 years	1	8
7–9 years	4	33
Greater than 9 years	6	50
Highest Level of Nursing Education		
Associate’s Degree (ADN)	2	17
Bachelor’s Degree (BSN)	4	33
Master’s Degree (MSN)	6	50
Years of RN Experience in the ED		
Less than 1 year	2	17
1–3 years	2	17
4–6 years	2	17
7–9 years	2	17
Greater than 9 years	4	32

**Table 2 healthcare-13-02386-t002:** Frequency table for pre- and post-Mini-Z Single Item Survey scores of participants who completed the project. Due to rounding errors, percentages may not equal 100%. (N = 6).

Variable	n	*%*
**Pre-MBMA MZSI**		
1: “I enjoy my work… no symptoms of burnout”	0	0
2: “I am under stress…don’t feel burned out.”	2	33
3: “I am definitely burning out…”	3	50
4: “The symptoms of burnout that I’m experiencing won’t go away…”	1	17
5: “I feel completely burned out…may need to seek help.”	0	0
**Post-MBMA MZSI**		
1: “I enjoy my work… no symptoms of burnout”	0	8
2: “I am under stress…don’t feel burned out.”	2	33
3: “I am definitely burning out…”	3	50
4: “The symptoms of burnout that I’m experiencing won’t go away…”	1	17
5: “I feel completely burned out…may need to seek help.”	0	0

**Table 3 healthcare-13-02386-t003:** Results of a Wilcoxon Signed-Rank Test conducted on the 12 participants who fully completed the pre-test Mini-Z Single Item Survey (MZSI), compared to the 12 participants who either fully completed the post-test MZSI (n = 6) or were accounted for through imputation when they fell to attrition (n = 6).

MZSI Pre-Intervention		MZSI Post-Intervention			
*M*	*SD*	*M*	*SD*	*Z*	*p*
2.75	0.87	2.78	0.81	−2.51	0.012

**Table 4 healthcare-13-02386-t004:** Results of a Wilcoxon Signed-Rank Test conducted on the six participants who completed the pre-test and post-test Nurses’ Well-Being Index Survey.

NWBI Pre-Intervention		NWBI Post-Intervention			
*M*	*SD*	*M*	*SD*	*Z*	*p*
3.5	1.47	3.00	2.07	−0.28	0.783

**Table 5 healthcare-13-02386-t005:** Results of a Wilcoxon Signed-Rank Test conducted on the 12 participants who fully completed the pre-test Nurses Well-Being Index Survey (NWBI), compared to the 12 participants who either fully completed the post-test NWBI (n = 6) or were accounted for through imputation when they fell to attrition (n = 6).

Pre-MBMA NWBI Score		Post-MBMA NWBI Score			
*M*	*SD*	*M*	*SD*	*Z*	*p*
2.92	1.68	3.08	1.56	−0.26	0.793

## Data Availability

The data that supports the findings will be available upon request.

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
