# Peer review of "A Mindfulness-Based Mobile Application’s Impact on Nurse Burnout Syndrome and Well-Being"

_healthcare, 2025, doi:10.3390/healthcare13192386_

Round 1
Reviewer 1 Report
Comments and Suggestions for Authors
Thank you for studying this important topic. I appreciate your study and the inclusion of qualitative results. Overall, your study will benefit from limiting the scope of conclusions and being forthright about the limitations of the study, especially the lack of findings and the small sample size. Presented in a clear and forthright way, I believe it is worth publishing.
Abstract
Strengths: the title is clear, and the abstract overall is clear and well written.
Areas to Improve: The abstract over-interprets individual improvements as "clinical improvements" and over interprets the data to conclude that nurses were more aware of their feelings at the second time point. Please remove the word clinical and adjust the language so as not to over-interpret the data.
Sentence to change: "Although statistical improvements were not observed, individual responses indicated clinical improvements in both burnout and wellbeing, alongside increased awareness of their feelings after engaging with the MBMA"
Introduction
Strengths: The introduction provides a compelling rationale for studying burnout in emergency department nurses. It integrates existing literature and mindfulness-based mobile application (MBMA) studies.
Areas to Improve: The introduction is solid. You could make the last sentence about the study goal more concise.
Methods
Strengths: The intervention, surveys, and structure of what you did are clearly described.
Areas to Improve: 50% of your post-test data is imputed. It would be good to see a primary analysis without imputed data (among responders) and the imputed analysis could serve as a secondary analysis.
You state "The Mini-Z Single Item Survey (MZSI), comprising of a single item with a 5-point Likert scale, was found to be an effective scale endorsed by the National Academy of Medicine due to its fast completion and comparable outcomes to other burnout instruments like the Maslach Burnout Inventory (r2 = 0.63-0.65, p < 0.05).32,33,34,"
Given the low correlation (0.63-0.65) I think it's a stretch to say that it is comparable -- perhaps you could say given the correlation with.
Also, the 0.64 number does not refer to correlation with the Malach Burnout Inventory as a whole, but rather with the emotional exhaustion component, so please clarify that.
From the source you cited "The single-item was correlated at r = 0.64 (p < 0.0001) with emotional exhaustion and the ANOVA yielded an R2 of 0.5 (p < 0.0001). Hence, in surveys of physicians where emotional exhaustion is the primary sub-scale of interest, a single-item measure of burnout may be used as an alternative to the Maslach Burnout Inventory in order to abbreviate survey material and potentially increase response rates among physicians"
Imputed data -- need to specify how data was imputed, it seems that you assigned the baseline score as the post-intervention score. This should be specified. I'm not sure if this is a valid approach, and I would think you should do the analysis with and without (and present both at minimum), but it should be specified the approach that was taken and repeated clearly in any figure and table legends that include imputed data.
Results
Strengths: Clear presentation of demographics. You mainly have a negative result, which in my mind is okay and worth reporting lack of findings. The qualitative aspects of the study are honestly very useful for those who wish to actually understand providers and design better tools.
Areas to Improve:
The before after results could be more clearly visualized with a plot that shows before and after boxplots for participants who did respond to both surveys (n=6).
I'd like to see both study measures visualized / in tables.
The sample size is very low, so a non-parametric test like Wilcoxon signed rank test may be more suitable than a t-test given that it's can be challenging to properly assess normality at such a low sample size. You could present both approaches.
The imputation rate is 50%, which is very high. I suggest to focus on the participants who completed the study as your primary analysis, and present the imputed version as a secondary approach, however I would defer to a statistician to advise on this point. You could present the analysis both ways, potentially with one in the supplement.
Figure 2: I am confused by the number of dots on the plot. You can use a scatter feature to make them more visible if they are overlapping or perhaps put a note.
4.5 Chi-Square Test of Independence
Demographics analysis: A sample size of six may be too low for a chi-square test. A Fisher's exact test might be a good fit.
4.6.1. Theme 1: Tools & Features -- confusing title
You mention that some participants identified technical barriers to using the app, but then you don't share their feedback until the end of the paragraph. You could rewrite this paragraph to be more concise.
Discussion
Strengths: You integrated data from the literature.
Areas to Improve: remove reference to clinical improvements, be mindful not to over-conclude past the scope of the study results. Be clear about the limited sample size. The part about nurses having increased awareness of their feelings is speculative and should be communicated as such. You may also want to focus more on "transferability and feasibility" as indicated from your study aim.
Specific points
You say in the introduction, "The purpose of this evidence-based practice and quality improvement (EBPQI) project was to explore the transferability, acceptability, and preliminary efficacy for nurses employed in the ED to use a once daily MBMA (Moodfit) tool on their feelings of burnout and well-being."
Then, in the discussion, you say you were not intending to assess efficacy. "
Although our results in this project did not show a significant reduction burnout or improve- ment in well-being when compared to baseline, our goal was not to determine efficacy; rather, we hoped to encourage and explore the use of a MBMA on its applicability and transferability of evidence into ED nursing practice." Please make these consistent and clear.
Lines 305-307: You mention you were able to capture "clinical improvements" in the nurses' well-being. This seems like a stretch. I suggest to delete the word clinical. You could say that while the sample size was low and study completion was low, several nurses exhibited improved scores and mentioned positive experiences, which deserves follow up, and suggests in line with the other literature you mentioned that these interventions can have a positive impact on people's lives, even if this study was not adequately powered to detect this. Overall, even non-significant improvements in scores were small, suggesting that there is a need for better interventions. The more honest you are about the results and limitations, the more interesting and valid the study becomes.
"In our EBPQI project, we observed a moderate and positive correlation between having poor well-being and worsening burnout. Although we can’t fully determine the correlation due to the small sample size, we theorize it could be attributed to the ED nurses becoming more self-aware of their feelings of burnout and poor wellbeing than before engaging with the MBMA."
This seems like a stretch and was also overstated in the abstract. Poor well being may be associated with burnout because burnout is an opposite experience to well-being. The lack of clear association in your pre-trial data doesn't necessarily mean that they became more aware (it might) and you didn't test this. You could also check to see whether the association exists at baseline when only including the study responders who had the association at study end.
Additionally, the ED nurses in our project reported near clinical impacts of burnout and poor being at baseline --> please fix this typo it should be well-being, not being :)
Overall please reduce the speculation throughout the discussion section, and be conservative around drawing conclusions from the limited dataset.
Limitations
The main limitations should be listed first. Your main limitation was sample size and lack of follow-up, not whether the nurses were from the emergency department. Relatedly, another limitation is that you imputed 50% of the data.
Suggest to delete, "we accept these limitations due to..." I would simply state that there were limitations that prevented determining the effectiveness of the intervention.
Regarding the app you used I actually experienced a personal improvement in stressful thoughts just looking it up and reading the description, though I liked it less using it. :) I'd be curious whether an app with a more modern user interface would perform better.
Author Response
|
RESPOSNES TO REVIEWER 1 COMMENTS
Abstract
Strengths: the title is clear, and the abstract overall is clear and well written. Sentence to change: "Although statistical improvements were not observed, individual responses indicated clinical improvements in both burnout and wellbeing, alongside increased awareness of their feelings after engaging with the MBMA"
|
|
Response 1: Thank you for your valuable feedback on our abstract. We appreciate your observation regarding the over-interpretation of results. After reflecting on your comments and updating our design methods, including the implementation of the Wilcoxon Signed Rank Test (in place of the Shapiro/Paired T-Test), we have made the necessary revisions.
Specifically, we have updated the abstract on page 1. Lines 17-18 now accurately present the outcomes of the Wilcoxon Signed Rank Test, focusing exclusively on participants who completed both pre- and post-tests, without any imputed data. Additionally, lines 18-19 address the secondary analysis of imputed data.
In lines 22-24, we have revised the language to better align with our participants' identification of the tool as helpful, based on qualitative results. We have clarified that while participants found the tool beneficial, we have omitted any claims of clinical improvements.
Please refer to the revisions highlighted in red within the manuscript for further details. Thank you again for your insights, which have strengthened our work.
|
|
Comments 2: Introduction Strengths: The introduction provides a compelling rationale for studying burnout in emergency department nurses. It integrates existing literature and mindfulness-based mobile application (MBMA) studies.
|
|
Response 2: Thank you for your feedback on the introduction. We understand that revising the last sentence can make the study's goal clearer, and we have modified lines lines 45-48 on page 2 of the manuscript to be more concise. Please refer to the updated in the manuscript, which is highlighted in red.
|
|
Comment 3: Methods: |
|
Point 1: Strengths: The intervention, surveys, and structure of what you did are clearly described. Areas to Improve: 50% of your post-test data is imputed. It would be good to see a primary analysis without imputed data (among responders) and the imputed analysis could serve as a secondary analysis.
|
|
Response 3: Thank you for your feedback on the methods section; we found much of it valuable. Regarding Point 1 (Areas to Improve), we have decided to include both primary and secondary analyses. We will use the Wilcoxon Signed-Rank Test to compare pre- and post-baseline differences among the six participants who completed all surveys. Additionally, we conducted secondary analysis on imputed data to account for the six participants who did not complete the post-surveys, thereby avoiding the potential bias of ignoring baseline levels of burnout and well-being. Updates have been made in the relevant sections of the manuscript to reflect these revised analysis techniques and results. See the page numbers and lines below, as well as the revisions highlighted in red text within the manuscript. · Page 3, lines 133-136: updating analysis techniques. · Page 5, lines 193-196: rationale for Wilcoxon Signed · Page 6, lines 217-218: revised the descriptive analysis of the pre-post-survey for n = 6 and 226-228, present the results from the primary analysis of the Wilcoxon Signed Rank Test. · Page 7, lines 237-242, provide the results for the Wilcoxon Signed Rank Rest for imputed data, along with Table 3. · Page 7, lines 250-251 and Table 4 (line 252). present the results for the Wilcoxon Signed Rank Test of the pre- and post NWBI scores of the six participants. · Page 8, lines 253-257 and Table 5 present the secondary analysis using the Wilcoxon Signed Rank Test and imputed data. Point 2. You state "The Mini-Z Single Item Survey (MZSI), comprising of a single item with a 5-point Likert scale, was found to be an effective scale endorsed by the National Academy of Medicine due to its fast completion and comparable outcomes to other burnout instruments like the Maslach Burnout Inventory (r2 = 0.63-0.65, p < 0.05).32,33,34," Given the low correlation (0.63-0.65) I think it's a stretch to say that it is comparable -- perhaps you could say given the correlation with. Response 4: Thank you for your valuable feedback regarding the validity/reliability of the MZSI, particularly in relation to its low correlation score and its use in measuring only specific aspects related to the burnout domain of emotional exhaustion. We have revised page 4, lines 146-149, to more accurately reflect the reliability and validity of the MZSI. Additionally, we have included in the manuscript the rationale for using a single-item burnout measure. The modifications are highlighted in red text within the manuscript.
Point 3. Imputed data -- need to specify how data was imputed, it seems that you assigned the baseline score as the post-intervention score. This should be specified. I'm not sure if this is a valid approach, and I would think you should do the analysis with and without (and present both at minimum), but it should be specified the approach that was taken and repeated clearly in any figure and table legends that include imputed data.
Response 5: Thank you for the additional explanation regarding the use of imputed data; we found it very helpful. To enhance the validity of our outcomes, we conducted both primary and secondary analyses, as discussed in Response 3. Additionally, the revisions specified in Response 3 clarify which data was imputed and which was not. We chose to retain the imputed data in our secondary analysis to ensure that we did not overlook baseline levels of burnout or well-being. While we acknowledge that imputed data may not be as valid as other approaches, our goal was to evaluate the preliminary efficacy, feasibility, and acceptability of the intervention. This approach sheds light on potential reasons for the high attrition rate, as many participants began with elevated levels of burnout or impaired well-being.
Comment 4: Results Strengths: Clear presentation of demographics. You mainly have a negative result, which in my mind is okay and worth reporting lack of findings. The qualitative aspects of the study are honestly very useful for those who wish to actually understand providers and design better tools.
Point 1: The before after results could be more clearly visualized with a plot that shows before and after boxplots for participants who did respond to both surveys (n=6). I'd like to see both study measures visualized / in tables. Response 6: Thank you for your feedback and valuable recommendations. We agree on the importance of clearly visualizing the results of both the primary and secondary analyses.
On page 6, lines 226-227, we present the outcomes of the Wilcoxon Signed Rank Test. Table 2 (on pages 6 and 7) provides a breakdown of the scores from the pre- and post-MZSI surveys and visualizes the descriptive analysis of the six participants.
Moving to page 7, Table 3 presents the Wilcoxon Signed Rank Test results for the imputed MZSI scores. Table 4, also on page 7, shows the Wilcoxon Signed Rank Test results for the pre- and post-NWBI scores from the primary analysis.
Finally, on page 8, Table 5 demonstrates the Wilcoxon Signed Rank Test results for the pre- and post-NWBI imputed scores.
Point 2: The sample size is very low, so a non-parametric test like Wilcoxon signed rank test may be more suitable than a t-test given that it's can be challenging to properly assess normality at such a low sample size. You could present both approaches.
Response 7: Thank you for clearly explaining the rationale for using the Wilcoxon Signed-Rank Test. Given the small sample size, we agree that this is a more appropriate test. As described in Response 3, the Wilcoxon Signed-Rank Test has been incorporated into the design, and all relevant areas have been highlighted in red text within the manuscript. · Page 3, lines 133-136: updating analysis techniques. · Page 5, lines 193-196: rationale for Wilcoxon Signed · Page 6, lines 217-218: revised the descriptive analysis of the pre-post-survey for n = 6 and 226-228, present the results from the primary analysis of the Wilcoxon Signed Rank Test. · Page 7, lines 237-242, provide the results for the Wilcoxon Signed Rank Rest for imputed data, along with Table 3. · Page 7, lines 250-251 and Table 4 (line 252). present the results for the Wilcoxon Signed Rank Test of the pre- and post NWBI scores of the six participants. · Page 8, lines 253-257 and Table 5 present the secondary analysis using the Wilcoxon Signed Rank Test and imputed data.
Point 3: The imputation rate is 50%, which is very high. I suggest to focus on the participants who completed the study as your primary analysis, and present the imputed version as a secondary approach, however I would defer to a statistician to advise on this point. You could present the analysis both ways, potentially with one in the supplement.
Response 8: Thank you for the rationale on data imputation and the recommendation to present both ways. We chose to present the data in both ways (primary and secondary analysis). All revisions are highlighted in red within the manuscript. · Page 6, lines 217-218: revised the descriptive analysis of the pre-post-survey for n = 6 and 226-228, present the results from the primary analysis of the Wilcoxon Signed Rank Test. · Page 7, lines 237-239 provide a rationale as to why an accurate assessment cannot be displayed definitively within each question criteria due to attrition when analyzing the primary data. · Page 7, lines 237-242, provide the results for the Wilcoxon Signed Rank Rest for imputed data, along with Table 3. · Page 7, lines 250-251 and Table 4 (line 252). present the results for the Wilcoxon Signed Rank Test of the pre- and post NWBI scores of the six participants. · Page 8, lines 254-257; Secondary analysis on NWBI imputed scores description. · Page 8, lines 253-257 and Table 5 present the secondary analysis using the Wilcoxon Signed Rank Test and imputed data.
Point 4. Figure 2: I am confused by the number of dots on the plot. You can use a scatter feature to make them more visible if they are overlapping or perhaps put a note.
Response 9: Thank you for your feedback and we can understand the confusion. We chose to place a note under the figure to foster clarity in the number of participants. The revision is highlighted in red within the manuscript on page 8, Figure 2.
Point 5. 4.5 Chi-Square Test of Independence
Response 10: Thank you for your feedback on the analysis of the relationship between education level and those who are MSN-prepared, using the Chi-Square Test of Independence. We agree and chose to eliminate using the Chi-Square Test of Independence due to its lack of reliability with a small sample size. While we did compile data using the Fisher's Exact Test, we did not find any relevant outcomes and chose to only discuss the outcomes within the discussion section, page 10, lines 352-365, with only revisions to verbiage in lines 352-357 (revisions can be found within the highlighted red text within the manuscript). The discussion highlights the findings that 5 out of 6 participants who completed the post-surveys were MSN-prepared. This connection aligns with previous literature suggesting that individuals with higher levels of education are more likely to be retained. This may lead to the hypothesis that those with higher education are more inclined to utilize individual-level resources, reinforcing the connections to existing literature that we previously presented and highlighting the need for further investigation. Additionally, to enhance clarity on the discussion of LOE and MSN prepared participants, who completed the post-intervention surveys, we decided to provide a more detailed description of the six participants on page 6, lines 209-215. Modifications are presented in red text within the manuscript.
Point 6. 4.6.1. Theme 1: Tools & Features -- confusing title
Response 11. We appreciate your feedback. To clarify the theme and title, we have updated the title to “Perceived Helpfulness of the MBMA Tools.” The revisions can be found on page 8, line 275, highlighted in red text. Additionally, we have made the paragraph more concise by starting with positive themes and concluding with challenges. The updated paragraph is located on page 9, lines 276-294, and is also highlighted in red text.
Comment 5: Discussion
Strengths: You integrated data from the literature. Areas to Improve: remove reference to clinical improvements, be mindful not to over-conclude past the scope of the study results. Be clear about the limited sample size. The part about nurses having increased awareness of their feelings is speculative and should be communicated as such. You may also want to focus more on "transferability and feasibility" as indicated from your study aim.
Point 1: You say in the introduction, "The purpose of this evidence-based practice and quality improvement (EBPQI) project was to explore the transferability, acceptability, and preliminary efficacy for nurses employed in the ED to use a once daily MBMA (Moodfit) tool on their feelings of burnout and well-being." Then, in the discussion, you say you were not intending to assess efficacy. " Although our results in this project did not show a significant reduction burnout or improve- ment in well-being when compared to baseline, our goal was not to determine efficacy; rather, we hoped to encourage and explore the use of a MBMA on its applicability and transferability of evidence into ED nursing practice." Please make these consistent and clear. Response 12: Thank you for your feedback. We agree that the clarity of the purpose and the discussion needs improvement. To ensure consistency and clarity, we revised the wording in lines 101-104 on page 3 (the last section of the background). Additionally, to align with this revised wording, we made a modification on page 9, lines 316-319 within the discussion section. The modifications are highlighted in red in the manuscript, as well as noted below. Revision (lines 101-104): “The purpose of this evidence-based practice and quality improvement (EBPQI) project was to explore the feasibility, acceptability, and preliminary efficacy, for ED nurses to use a once daily MBMA (MoodfitTM) tool on their feelings of burnout and well-being over a four week time.” Revision (lines 316-319 ): “Although our results in this project did not show a significant reduction in burnout or improvement in well-being when compared to baseline, our goal was to determine feasibility, acceptability, and preliminary efficacy to explore the MBMA utility in ED nursing practice.”
Point 2: Lines 305-307: You mention you were able to capture "clinical improvements" in the nurses' well-being. This seems like a stretch. I suggest to delete the word clinical. You could say that while the sample size was low and study completion was low, several nurses exhibited improved scores and mentioned positive experiences, which deserves follow up, and suggests in line with the other literature you mentioned that these interventions can have a positive impact on people's lives, even if this study was not adequately powered to detect this. Overall, even non-significant improvements in scores were small, suggesting that there is a need for better interventions. The more honest you are about the results and limitations, the more interesting and valid the study becomes. "In our EBPQI project, we observed a moderate and positive correlation between having poor well-being and worsening burnout. Although we can’t fully determine the correlation due to the small sample size, we theorize it could be attributed to the ED nurses becoming more self-aware of their feelings of burnout and poor wellbeing than before engaging with the MBMA." Additionally, the ED nurses in our project reported near clinical impacts of burnout and poor being at baseline --> please fix this typo it should be well-being, not being :) Response 13: Thank you for your feedback. We found it meaningful and made revisions based on the use of the Wilcoxon Signed Rank Test. To reduce speculation, we have removed the term “clinical improvement” from the manuscript. Additionally, lines 316-326 (pages 9-10) have been modified to reflect the non-significant findings while still capturing certain improvements and connecting to qualitative data and disengagement. Please see the revisions highlighted in red within the manuscript.
Furthermore, we have added a supplementary plot (in the supplementary material & is noted in line 323) to display the changes in pre- and post-scores among the six participants. This addition aims to provide transparency regarding the participants who demonstrated improved scores following the intervention.
Comment 6: Limitations
The main limitations should be listed first. Your main limitation was sample size and lack of follow-up, not whether the nurses were from the emergency department. Relatedly, another limitation is that you imputed 50% of the data. Regarding the app you used I actually experienced a personal improvement in stressful thoughts just looking it up and reading the description, though I liked it less using it. :) I'd be curious whether an app with a more modern user interface would perform better. Response 14: Thank you for your feedback and valuable input on the limitations section. We agree that there are improvements to be made in the organization, structure, and clarity of this section. We have revised the entire limitations section on page 11, lines 376-394, and we look forward to your feedback. The revisions begin with the main limitation (small sample size), thoroughly discuss the use of the imputed method, and conclude by connecting why these findings are still relevant.
|
|
5. Additional clarifications
Response 15: Additional revisions were made to improve the conciseness of various areas. Identification and rationale of revisions are presented below.
Page 1, lines 21-22; Page 8, lines 272-274; and Page 9, lines 395-313. The theme titles were updated to better reflect the participants experiences and clarity with the titles ( to further improve based on Comment 4, Point 6 and confusion on title).
To better align with the revisions, we improved the conciseness of the discussion and conclusion, reduced speculation, and enhanced the wording. Additional improvements made within the Discussion section; please refer to the highlighted red text on page 10 with lines 342-343 and 368-374. Modifications can also be found in the Conclusion on page 11, specifically in lines 398-399 and 403-406.
|
Reviewer 2 Report
Comments and Suggestions for Authors
Title: I believe the title needs to be revised as it's not the impact of the application, but it’s the impact of the intervention that is delivered through a mobile-based app to relieve the symptoms of burnout and enhance well-being among nurses from ED.
Comment: Based upon the review of the manuscript, I see the major thing that needs to be considered here is the methodology. Based upon this study, if we conclude that this mobile-app-based intervention was not effective in relieving the symptoms of burnout or improving well-being, we might not be accurate in our conclusion because there are too many limitations of the methodology, which are listed as follows
- The measures used to assess burnout and well-being are based upon a single item, and the statistical method used becomes less accurate in their estimation when items used are based on a single item.
- The sample size is limited to six participants who completed both pre-test and post-test
- The duration of both pre-and post is a total of four weeks, and the impacts of psychological interventions on variables such as well-being and burn-out, which also deteriorate over a long period of time, as well do not recover over such short period of time when also measured by just single item indicator.
Recommendation: I recommend that this research paper should only be based upon qualitative data. The sample size match to meet the requirements of a qualitative research. Authors can elaborate more on the qualitative findings that provides more insight into user experiences, and these findings could be used to improve the self-administered mobile-based interventions to support well-being among vulnerable populations.
Author Response
|
Response to Reviewer 2 Comments |
||
|
1. Summary |
|
|
|
Thank you very much for taking the time to review this manuscript. Please find the detailed responses below and the corresponding revisions/corrections highlighted or in red text as requested in the re-submitted files.
|
||
|
2. Questions for General Evaluation |
Reviewer’s Evaluation |
Response and Revisions |
|
Does the introduction provide sufficient background and include all relevant references? |
Yes/Can be improved/Must be improved/Not applicable |
Thank you for the feedback demonstrating a strong introduction. Revisions to the Introduction were made based on Reviewers 1 feedback. Please see the highlighted red sections within the manuscript. |
|
Are all the cited references relevant to the research? |
Yes/Can be improved/Must be improved/Not applicable |
No comments were present regarding the need for revisions to this question. |
|
Is the research design appropriate? |
Yes/Can be improved/Must be improved/Not applicable |
A response will be noted in the point-by-point response section: Methods. Revisions were made. All revisions are noted in red within the manuscript text. Major revisions included the use of the Wilcoxon Ranked Signed Test, elimination of the Chi-Square. |
|
Are the methods adequately described? |
Yes/Can be improved/Must be improved/Not applicable |
A response will be noted in the point-by-point response section: Methods. Revisions were made. Additional rationales were added for the use of imputed data and a primary and secondary analysis. All revisions are highlighted in red text within the manuscript.
|
|
Are the results clearly presented? |
Yes/Can be improved/Must be improved/Not applicable |
A response will be noted in the point-by-point response section: Results. Revisions were made. All revisions are highlighted in red text within the manuscript.
|
|
Are the conclusions supported by the results? |
Yes/Can be improved/Must be improved/Not applicable |
A response will be noted in the point-by-point response section: Results. Revisions were made. All revisions are highlighted in red text within the manuscript.
|
|
3. Point-by-point response to Comments and Suggestions for Authors |
||
|
Comments 1: Title I believe the title needs to be revised as it's not the impact of the application, but it’s the impact of the intervention that is delivered through a mobile-based app to relieve the symptoms of burnout and enhance well-being among nurses from ED
Response 1: Thank you for your insightful comment regarding the title of our manuscript. We appreciate your suggestion to revise it for greater clarity. After careful consideration, we have chosen to retain the original title as it accurately reflects the focus of our project. While we understand that the intervention is delivered through the mobile application, we feel the current title effectively conveys both the intervention (mindfulness-based approach) and its delivery method (mobile application), while keeping the emphasis on the outcomes related to burnout and well-being. Additionally, with various mobile applications that deliver mindfulness-based interventions (Calm or Headspace), as well as various strategies to foster mindfulness, we felt it was important to highlight the specific medium through which this intervention is administered.
We hope this explanation clarifies our choice, and we are happy to further discuss or refine any aspects as needed.
Comment 2: Based upon the review of the manuscript, I see the major thing that needs to be considered here is the methodology. Based upon this study, if we conclude that this mobile-app-based intervention was not effective in relieving the symptoms of burnout or improving well-being, we might not be accurate in our conclusion because there are too many limitations of the methodology, which are listed as follows The measures used to assess burnout and well-being are based upon a single item, and the statistical method used becomes less accurate in their estimation when items used are based on a single item. The sample size is limited to six participants who completed both pre-test and post-test The duration of both pre-and post is a total of four weeks, and the impacts of psychological interventions on variables such as well-being and burn-out, which also deteriorate over a long period of time, as well do not recover over such short period of time when also measured by just single item indicator.
Response 2: Thank you for your thoughtful and constructive feedback. We appreciate your detailed review and recognize the concerns you have raised regarding the methodology, particularly around the sample size, measurement approach, and the short duration of the intervention. We agree that these factors can limit the robustness of conclusions, and we have made revisions to address these issues.
Revisions are indicated in red within the Materials and Methods and Results sections, specifically on pages 3 to 9 of the manuscript.
Comment 3: Recommendation: I recommend that this research paper should only be based upon qualitative data. The sample size match to meet the requirements of a qualitative research. Authors can elaborate more on the qualitative findings that provides more insight into user experiences, and these findings could be used to improve the self-administered mobile-based interventions to support well-being among vulnerable populations.
Response 3: Thank you for your constructive recommendation and thoughtful feedback. We truly appreciate your perspective on the potential value of focusing solely on qualitative data. After carefully reflecting on your suggestion, we agree that qualitative findings can offer valuable insights into user experiences. However, we believe that the combination of qualitative and quantitative data in our study strengthens the overall conclusions. The qualitative data provides rich, contextual insights into the experiences of participants, but we feel that quantitative results are essential to support and validate these findings. Without the quantitative analysis, the qualitative results might not fully capture the extent of the intervention's impact, especially given the limited sample size. In response to your suggestion, we have ensured that the qualitative findings are elaborated upon more clearly in the revised manuscript, particularly in relation to the user experiences. Additionally, we have addressed the revisions based on Reviewer 1's feedback (highlighted in red within the manuscript) to strengthen both the qualitative and quantitative analyses. We hope that, with these revisions, the manuscript is sufficient as it currently stands.
Thank you again for your thoughtful suggestions, and we welcome any further feedback you may have.
|
||
Round 2
Reviewer 1 Report
Comments and Suggestions for Authors
Great job implementing the suggesting changes.
Abstract
Line 23-24: Themes from qualitative responses included perceived helpfulness of MBMA tools, perceived usefulness, and lack of time (this is unclear, especially the lack of time part, can you rephrase?)
Lines 25-27: I'm not sure this is a complete sentence. Please make more clear.
Conclusion of Abstract
Great job
Introduction
Lines 106-109: Better!
Materials and Methods
This section was not appropriately updated to reflect the new approach and specify the imputation process.
Line 138: Data is a plural word, this should say were analyzed not was analyzed.
Line 138-140: This is not clear. You do not describe what primary data entails. You need to clearly describe all the statistical analyses you performed in the Methods section. Someone needs to walk away knowing exactly what was done, and which participants were included and excluded from each analysis. The rationale for why you chose the test should go here too I believe.
Line 152: Suggest to delete the word therefore, it is not used correctly in this context.
Line 172: Suggest to change the word "Since" to "Given that" or something similar.
Results
Thank you for updating the results section and adding tables to clearly show before and after scores. It might be easier to get a sense of the data with box or line plots. Line plots would allow us to see how each participant did over time.
Line 198-199: Please make clear as to what was done on which participants. Suggest: "We ran a Wilcoxon Signed Rank test to compare preliminary differences in burnout and wellbeing among the six participants who completed the study."
Lines 213-219: This is slightly hard to read and grasp. It might make more sense to update the table with the results of the 6 and to move the current demographics table for the 12 to the supplement (or vice versa). You can look up whether it's more appropriate to include the analyzed set or the full set in the main demographics table.
Lines 221-224: This can be worded more clearly. "Among participants who completed the study (n = 6), 67% reported emotional exhaustion at baseline. Specifically, four participants responded they had “one or more symptoms of burnout,” or “having symptoms that will not go away.” Two participants did not have burnout at baseline.
(Also you can clarify it's emotional exhaustion)
If there were participants who had non-significant improvements in their score you can report that here since you touch on it in the abstract and discussion but don't report the data to back it up.
Line 231: "Yet individual scores varied between participants" is unclear
When primarily analyzing responses from participants who completed both the pre- and post-surveys completely (n = 6),...
Lines 241-243: This is grammatically incorrect
Tables: All tables need more detailed legends.
Table 2: Table 2 is not an great way to get a sense of the scores as we can't tell whether individual participants improved as you say that they did. I suggest replacing this with a plot that shows each participant before and after so we can track how each individual did. A visual would be more clear.
4.3 Nurses’ Well-Being Index Survey
Lines 249-254: This is dense and confusing.
Lines 256-261: This is unclear and confusing. Be concise and specific.
Figure 1: The note is not adequate and does not help me understand what is going on. Please update the plot so we can see each dot or describe every dot that is obscured in a clear concise way. This needs to fixed appropriately before it can be published.
Finally, I recommend you call every one of the six participants who did not complete the post-intervention survey and ask if they can complete the perceived helpfulness survey, and then include those results.
Discussion
Lines 324-325: I do not understand what you mean by their sense of burnout and well-being and as mentioned in the prior report I don't agree with your conclusion that they had an increased awareness of their state, that is not supported by data, though it could be a hypothesis.
Lines 327-330: This can be worded better. Start off by saying that participants who were retained found the reminder helpful (this should be in the results) and then you can make the suggestion lightly.
Lines 330-332: The study you cite doesn't relate to the point you made in the prior sentence about reminders being helpful. This doesn't flow in a clear way.
Lines 346-347: Suggest to change remaining to completing to be more clear.
Lines 356-361: Clarify that the sample size was small here. "However, the sample size was very small."
Lines 390-391: Delete "and provide a prediction of preliminary outcomes." This doesn't make sense.
Lines 391-395: Make this more clear and concise.
Lines 402-405: Need to improve.
Suggest: Although the examined mindfulness-based mobile application did not generate statistically significant improvements in well-being or burnout, open-ended responses about the MBMA, and non-significant improvements in XX nurses align with findings from other studies that suggest that MBMA's can ....
407-410: This seems speculative to make a proposal and unrelated to what you mentioned prior. Suggest to fully delete this sentence.
Limitations
Much better!
Lines 383-385: I'm not convinced that running an analysis on people who didn't respond accounts for the small sample size.
Given that one of your references did not relate to your point in the prior iteration I suggest you check each reference individually. I did not check every single one for you.
Author Response
For research article: A Mindfulness-Based Mobile Applications’ Impact on Nurse Burnout Syndrome and Well-being.
Response to Reviewer 1 Comments: Review Report Round 2
|
1. Summary |
|
|
|
Thank you very much for taking the time to review this manuscript. Please find detailed responses below and the corresponding revisions/corrections highlighted in yellow for the second round. The red text is the revisions from the first round.
|
||
|
2. Questions for General Evaluation |
Reviewer’s Evaluation |
Response and Revisions |
|
Does the introduction provide sufficient background and include all relevant references? |
Yes/Can be improved/Must be improved/Not applicable |
No comments were present regarding the need for revisions to this question, however the abstract was updated to reflect updated revisions. Responses are noted below. |
|
Are all the cited references relevant to the research? |
Yes/Can be improved/Must be improved/Not applicable |
No comments were present regarding the need for revisions to this question, however, references were double checked to ensure relevance. |
|
Is the research design appropriate? |
Yes/Can be improved/Must be improved/Not applicable |
No comments were present regarding the need for revisions to this question. |
|
Are the methods adequately described? |
Yes/Can be improved/Must be improved/Not applicable |
A response will be noted in the point-by-point response section: Methods. Revisions were made.
|
|
Are the results clearly presented? |
Yes/Can be improved/Must be improved/Not applicable |
A response will be noted in the point-by-point response section: Results. Revisions were made. |
|
Are the conclusions supported by the results? |
Yes/Can be improved/Must be improved/Not applicable |
A response will be noted in the point-by-point response section: Results. Revisions were made.
|
|
Point-by-point response to Comments and Suggestions for Authors |
|
Comments and Suggestions for Authors: Great Job implementing the suggesting changes. Abstract Comments 1: Line 23-24: Themes from qualitative responses included perceived helpfulness of MBMA tools, perceived usefulness, and lack of time (this is unclear, especially the lack of time part, can you rephrase?) Lines 25-27: I'm not sure this is a complete sentence. Please make more clear. Conclusion of Abstract Great job
|
|
Response 1: Thank you for your valuable feedback on our abstract. We appreciate your observation regarding the clarity of how the themes are displayed, as well as the writing mechanics of lines 25-27. After reflecting on your comments, we have made the following revisions to improve clarity:
Point 1. Specifically, in lines 26 we have revised the language to address the clarity of one of the qualitative themes “lack of time” with the changes highlighted in yellow.
Point 2. Specifically, in lines 26-28 language was improved to be grammatically correct. The changes are highlighted in yellow. Thank you again for your insights, which have strengthened our work.
|
|
Comments 2: Materials/Methods Line 138: Data is a plural word, this should say were analyzed not was analyzed. Line 138-140: This is not clear. You do not describe what primary data entails. You need to clearly describe all the statistical analyses you performed in the Methods section. Someone needs to walk away knowing exactly what was done, and which participants were included and excluded from each analysis. The rationale for why you chose the test should go here too I believe. Line 152: Suggest to delete the word therefore, it is not used correctly in this context. Line 172: Suggest to change the word "Since" to "Given that" or something similar. |
|
Response 2: Thank you for your feedback on the materials and methods sections. We reflected on your comments and revised them appropriately. See the point by point below for each line. Please refer to the updates in the manuscript, highlighted in yellow.
Point 1. We agree with this comment and made appropriate revisions to identify the use of the imputation process. Revisions can be seen in lines 144-146. Point 2. Thank you for identifying this grammatical error. We agree with the feedback and revised “was” to “were” to align with the plural form of “data”. See Line 142. Point 3. Thank you for the feedback on providing a more detailed description of the data analysis techniques. After reflecting on the comments, we did make additional revisions identifying the rational for keeping all 12 participant demographics (lines 136-139), use of the imputation process (lines 144-146), and identified the use of the Pearson Correlation test with rationale (lines 148-151). Revisions to the language surrounding the recommended change were included to foster more clarity. See revisions highlighted in yellow. Point 4. Thank you for your feedback, and the word “therefore” was deleted with a minor revision to ensure clarity of the sentence. See lines 160-161 within the manuscript, highlighted in yellow. Point 5. Thank you for your feedback, and the recommended revision was made to begin the sentence with “Given that.” Please see revision in line 181 within the manuscript, highlighted in yellow.
|
|
Comment 3: Results Thank you for updating the results section and adding tables to clearly show before and after scores. It might be easier to get a sense of the data with box or line plots. Line plots would allow us to see how each participant did over time.
Line 198-199: Please make clear as to what was done on which participants. Suggest: "We ran a Wilcoxon Signed Rank test to compare preliminary differences in burnout and wellbeing among the six participants who completed the study."
Lines 213-219: This is slightly hard to read and grasp. It might make more sense to update the table with the results of the 6 and to move the current demographics table for the 12 to the supplement (or vice versa). You can look up whether it's more appropriate to include the analyzed set or the full set in the main demographics table.
Lines 221-224: This can be worded more clearly. "Among participants who completed the study (n = 6), 67% reported emotional exhaustion at baseline. Specifically, four participants responded they had “one or more symptoms of burnout,” or “having symptoms that will not go away.” Two participants did not have burnout at baseline. (Also you can clarify it's emotional exhaustion) If there were participants who had non-significant improvements in their score you can report that here since you touch on it in the abstract and discussion but don't report the data to back it up.
Line 231: "Yet individual scores varied between participants" is unclear When primarily analyzing responses from participants who completed both the pre- and post-surveys completely (n = 6),...
Table 2: Table 2 is not an great way to get a sense of the scores as we can't tell whether individual participants improved as you say that they did. I suggest replacing this with a plot that shows each participant before and after so we can track how each individual did. A visual would be more clear.
4.3 Nurses’ Well-Being Index Survey Lines 249-254: This is dense and confusing. Lines 256-261: This is unclear and confusing. Be concise and specific.
Figure 1: The note is not adequate and does not help me understand what is going on. Please update the plot so we can see each dot or describe every dot that is obscured in a clear concise way. This needs to fixed appropriately before it can be published.
Finally, I recommend you call every one of the six participants who did not complete the post-intervention survey and ask if they can complete the perceived helpfulness survey, and then include those results.
|
|
Response 3 Point 1. We appreciate the time taken to thoroughly review our manuscript. We agree with the recommended wording change in the first paragraph of the Results section distinguishing what was done on which participants to foster more clarity. Revisions have been made in lines 203-210 highlighted in yellow within the manuscript. Minor adjustments were also incorporated to improve clarity and language, ensuring alignment with the suggested change.
Point 2. Thank you for your helpful feedback regarding the clarity of the participant characteristics (n = 6). We appreciate the suggestion of including an additional table for the six participants who completed both the pre- and post-intervention surveys. While we carefully considered this option, we were also mindful of limiting the total number of tables and figures to avoid crowding the manuscript. However, we have revised this section to improve clarity and readability. Revisions are highlighted in yellow within lines 214-220. /We have also included a section (above 4.2) identifying the primary characteristics of the six participants who completed the post surveys (lines 223-229). The section specifically highlights the characteristics of the retained participants, and we believe the narrative presentation allows for sufficient comparison within the discussion section. If you still feel that an additional table would substantially improve the manuscript, we are open to including it in the revision or as a supplementary document.
Point 3. Thank you for this helpful suggestion. We agree that the wording in this section could be made clearer and have revised lines 231-240 to improve clarity in how the data are presented, particularly regarding emotional exhaustion. Table 2 was also moved and aligned with the descriptive data for the MZSI (line 242). See revisions within the manuscript, highlighted in yellow.
Point 4. After reflecting on the comment regarding the clarity of the primary analysis using the Wilcoxon Signed Rank Test—specifically the phrasing “yet individual scores varied between participants”—we agreed with the suggested revision. Changes have been made in lines 244-248, highlighted in yellow in the manuscript. Additionally, a Profile Plot was added (Figure 1/ line 252) to visualize the changes in pre- and post- test scores of the six participants who completed the MZSI.
Point 5. Thank you for noting this grammatical error (within the original lines 241-243). We have revised the sentence for correctness and clarity. The updated wording can be found in lines 259-261, highlighted in yellow in the manuscript.
Point 6. Thank you for the recommendations regarding the tables and legends, and your attention to detail. We chose not to use the journal’s editorial service to revise the tables and figures due to associated costs, however, we have reviewed other publications within MDPI and updated the legends for all tables and figures to improve clarity and consistency. The revisions for all tables and figures are highlighted in yellow within the manuscript for review. However, if the current revisions are not sufficient, we would consider submitting them to professional editorial services. See all updated legends for all tables and figures, highlighted in yellow within the manuscript.
Point 7. We appreciate the reviewer’s suggestion. To better illustrate individual changes, we will include a line plot after the presentation of the NWBI results. This plot will display pre- and post-intervention scores for each of the six participants who completed all surveys, allowing readers to track changes in both the MZSI and NWBI scores visually. While this was originally submitted as supplemental material, it will now be included within the manuscript. Please see the added figures (Figure 1 and Figure 2) in lines 252 and 283 highlighted in yellow within the manuscript.
Point 8. We appreciate the reviewer’s comment regarding the unclear and confusing wording in lines 249–254 and 256–264. Upon review, we agree that revisions were needed. We also identified a typo in the first sentence of Section 4.3 (“When primarily analyzing participants who completed both the pre- and post-surveys (n = 6)…), this data reflects the baseline data amongst the 12 participants, which makes it inaccurate and leading to confusion for the remaining section. This sentence has been restructured and the section reorganized for clarity and conciseness. Please see the yellow highlighted revisions in lines 271-278, 285-287, 290-294. Additionally, a Profile Plot (Figure 2) was added to visualize the NWBI scores for the participants who completed the pre- and post-tests.
Point 9. Thank you for your careful and detailed review, which has helped strengthen the manuscript and maintain professional standards. We understand the importance of clearly presenting the results of the Pearson Correlation between burnout and well-being scores. This analysis was initially suggested by a consulting statistician; however, all other data analysis methods were completed by the principal investigators using Intellectus. To enhance the clarity of the original Figure 1, we re-analyzed the data in Intellectus using alpha levels of .5 and .1-with outcomes being the same. The overall outcomes were consistent—comparing the Pre MZSI and Pre NWBI scores showed a low correlation among the six participants, while the Post MZSI and Post NWBI scores showed a high correlation. However, the r-values differed slightly, likely due to differences between manual calculation by the statistician and the Intellectus software. We confirmed that the correct scores were inputted, and the regression lines visually match the expected relationships. The Intellectus analysis has been included as supplementary material. To further promote clarity, we have added two separate scatter plots (Figures 3 & 4), one for the pre-intervention data and one for the post-intervention data along with descriptive legends. Figure numbers changed, due to an addition reflecting before and after scores of the six participants on all surveys. All revisions are highlighted in yellow within lines 295-309 of the manuscript, as well as updated and highlighted within the abstract. We hope these changes adequately address your concerns and welcome any additional feedback or suggestions.
Point 10. We appreciate the suggestion to reach out to all participants who completed the pre- and post-tests to complete the perceived helpfulness survey, as this could provide further strength for the findings. However, perceived helpfulness was identified as a qualitative theme rather than a formal measurement tool in the project design. Additionally, the project was IRB approved as an anonymous and confidential project, with all contact information deleted upon completion. As such, follow-up contact with participants was not possible under the approved protocol.
|
|
Comment 4 Discussion Lines 324-325: I do not understand what you mean by their sense of burnout and well-being and as mentioned in the prior report I don't agree with your conclusion that they had an increased awareness of their state, that is not supported by data, though it could be a hypothesis. Lines 327-330: This can be worded better. Start off by saying that participants who were retained found the reminder helpful (this should be in the results) and then you can make the suggestion lightly. Lines 330-332: The study you cite doesn't relate to the point you made in the prior sentence about reminders being helpful. This doesn't flow in a clear way. Lines 346-347: Suggest to change remaining to completing to be more clear. Lines 356-361: Clarify that the sample size was small here. "However, the sample size was very small." Lines 383-385: I’m not convinced that running an alyasis on people who didn’t respon accounts for the small sample size. Lines 390-391: Delete "and provide a prediction of preliminary outcomes." This doesn't make sense. Lines 391-395: Make this more clear and concise. Lines 402-405: Need to improve. Suggest: Although the examined mindfulness-based mobile application did not generate statistically significant improvements in well-being or burnout, open-ended responses about the MBMA, and non-significant improvements in XX nurses align with findings from other studies that suggest that MBMA's can .... 407-410: This seems speculative to make a proposal and unrelated to what you mentioned prior. Suggest to fully delete this sentence.
|
|
Response 4 Thank you for your feedback on the Discussions, Limitations, and Conclusion sections. We reflected on your comments and revised them appropriately. See the point by point below for each line. Please refer to the updates in the manuscript, highlighted in yellow.
Point 1. Thank you for emphasizing this point, regarding the original Lines 324-325 and the discussion on whether increased awareness is supported by data. As we reflected on this comment, we researched additional data to suggest a correlation between mindfulness and subjective well-being. To strengthen the discussion, we added lines 379-382. Additionally, we did revise the wording in lines 361-374 to align more with a hypothesis, and changed some wording in the discussion (line 389). Updated text can be found in the manuscript highlighted in yellow.
Point 2. Thank you for this helpful suggestion. This section (original lines 327-330) has been revised to better align with your feedback. We now begin by highlighting that participants who were retained found the reminder helpful and presented the suggestion in a more balanced and subtle manner. Please see the revised text in Lines 367-373 highlighted in yellow.
Point 3. Thank you for pointing this out. We understand that the cited article may have appeared misaligned with the preceding sentences. Upon reviewing the research more carefully, we revised this section to improve clarity and better articulate how the article supports the stated hypothesis. Additionally, we incorporated the previously discussed article linking mindfulness and subjective well-being to further demonstrate that participants who engaged with the intervention may have experienced improvements. These revisions can be found on Lines 373–377 and 379–382 and are highlighted in yellow within the manuscript.
Point 4. Thank you for your recommendation regarding the wording for clarity. In response, we revised the original phrasing in line 395, replacing “remaining” with retained” to improve the flow and precision of the sentence.
Point 5. Thank you and we agree this project has many limitations with the main one being the small sample size and agree with emphasizing this prior to our explanation/hypothesis. Please see lines 406-407 with the recommended revision highlighted in yellow.
Point 6. Thank you for recognizing the strong revisions within the limitations section. While we understand that using the imputation process cannot replace actual measurements, it is data analytic strategy to integrate to avoid discounting pre- data. To address the comment, we did incorporate additional verbiage on the imputation process within the limitations section. Please see lines 436-438, highlighted in yellow within the manuscript. Furthermore, to address the feedback for liens 390-391, “provide a prediction for preliminary findings” was deleted and discussion on imputation was revised.
Point 7. Thank you for pointing out the need to be more concise in our writing. To address this feedback, we have revised the original Lines 387-391 and 391-395. Please see the revised limitation section, specifically lines 440-445, highlighted in yellow within the manuscript.
Point 8. Thank you for your feedback on our conclusion section and for highlighting the importance of avoiding over speculation. In response, we have revised the wording to incorporate some of your recommended languages and to better reflect the updated evidence presented in the project. These changes were made to ensure alignment with previous revisions and to maintain a balanced and appropriate tone in our conclusions. Please see the revisions within lines 451-454 and 456-458 highlighted in yellow within the manuscript.
Additional Revisions: To ensure we are submitting a strong response, we did review the article and references in their entirety. Additionally, we revised wording and the order of presentation in lines 412-416. All revisions for round two are highlighted in yellow within the manuscript.
We sincerely appreciate the time and care you have taken in providing thorough and constructive feedback throughout the review process. Your insights have greatly contributed to strengthening our manuscript, and we look forward to the opportunity to progress and share this research further.
|
